# Exploring caregivers' experiences of Kangaroo Mother Care in Bangladesh: A descriptive qualitative study

**Johanna Sjömar**[1]☯*, **Hedda Ottesen**[1]☯, **Goutum Banik**[2], **Ahmed Ehsanur Rahman**[2], **Ylva Thernström Blomqvist**[1], **Syed Moshfiqur Rahman**[1], **Mats Målqvist**[1]

**1** Department of Women's and Children's Health, Uppsala University, Sweden, **2** Maternal and Child Health Division, International Centre for Diarrhoeal Research, Bangladesh

☯ These authors contributed equally to this work.
* johanna.sjomar@kbh.uu.se

**Data Availability Statement:** All relevant data are within the paper and its Supporting Information files. Raw data (i.e. transcripts and translations) cannot be shared publicly due to assurances of

## Abstract

### Background

Kangaroo Mother Care (KMC) is an evidence-based intervention recommended by the World Health Organization (WHO) to reduce preterm mortality and morbidity. The aim of this study was to explore caregivers' experiences of providing KMC in hospital settings and after continuation at home in Bangladesh in order to assess enablers and barriers to optimal implementation.

### Methods

Interviews with fifteen caregivers were conducted using an interview guide with semi-structured questions in August 2019 and March 2020. Convenience sampling was used to select hospitals and participants for the study. The inclusion criteria were being a caregiver currently performing KMC in the hospital or having been discharged one week earlier from the KMC ward. The interviews were audio recorded, transcribed verbatim, and translated. The data were analyzed using thematic analysis with an inductive approach.

### Results

Three themes were identified as regards the caregivers' experiences of providing KMC: *conducive conditions*, *an empowering process*, and *suboptimal implementation*. The results showed that there are supporting circumstances for caregivers performing KMC in Bangladesh, including social support structures and positive attitudes to the method of care. It also appeared that the caregivers felt strengthened in their roles as caregivers by learning and performing KMC. However, the implementation of KMC was suboptimal due to late initiation of KMC, difficulties with keeping the baby skin-to-skin, and pain after cesarean section hampering skin-to-skin practice.

confidentiality given to participants at the time of consenting. icddr,b is the local collaborator and ethics approval was obtained from the Ethical Review Committee at icddr,b in Dhaka, Bangladesh. Because of the statutory requirements, internal data policies and regulations existing in the collaborating bodies along with the over-arching General Data Protection Regulation (GDPR), the data must be stored in institutional repository (storage platforms) and cannot be made directly accessible without a review of the request for access to data. Therefore, the data can be accessed only upon formal request that details the purpose of such request. The request will then be processed by the Data Repository Committee (DRC) at icddrb. Any such request should be directed to the principal investigator Dr. Ahmed Ehsanur Rahman (ehsanur@icddrb.org) or to the Mr. M A Salam Khan, IRB coordinator secretariat (salamk@icddrb.org).

**Funding:** This project has been made possible by grants from Födelsefonden (JS), Gillbergska (JS) and Minor Field Study (HO). The funders had no role in the study design, data collection and analysis, descision to publish, or preparation of the manuscript.

**Competing interests:** The authors have declared that no competing interests exist.

## Conclusions

The social and cultural conditions for the caregivers to perform KMC as well as the empowerment the parents felt in their roles as caregivers when performing KMC are facilitating factors for this method of care. Initial separation and late initiation of KMC, as well as disregard for the mothers' needs for care and support, were barriers to optimal practice leading to missed opportunities. These facilitators and barriers need to be addressed in order to succeed in scaling up the national KMC program.

## Introduction

Every year, 5.2 million children die before their fifth birthday [1]. Almost half of them die during the first month of life due to preventable conditions. Preterm birth with complications is the leading direct cause of 35% of the world's 2.4 million neonatal deaths each year [2]. Low birth weight (LBW) babies are at higher risk of morbidity such as infections, growth retardation, and developmental delay, and also have a higher risk of dying during infancy or childhood than other babies [3].Without concerted efforts to address these issues, it will be difficult to achieve the 2030 Sustainable Development Goal of reducing neonatal mortality to at least as low as 12 per 1000 live births [4].

The World Health Organization (WHO) recommends Kangaroo Mother Care (KMC) for preterm and low birth weight (LBW) babies with a birth weight of 2000 g or less [3]. The core of KMC is early, continuous, and prolonged skin-to-skin contact between the mother or other caregiver and the newborn. It also includes exclusive and frequent feeding with breastmilk (if possible) or giving breastmilk and early discharge from hospital and follow-up [5]. KMC is conceptualized as a "total health care strategy" [6] which is applied within a supportive environment where the mother of an LBW or premature baby is supported by healthcare workers and family members both in the healthcare facility and in the community once at home.

KMC benefits the newborn as well as the caregiver and family. It reduces the length of hospital stay after birth, promotes weight gain, breastfeeding, and neurobehavioral development, and helps maintain the baby's blood glucose levels. It also reduces the risk of hypothermia, infections, pain response, and neonatal mortality [5, 7, 8]. KMC facilitates the attachment between the newborn and the caregiver, reduces the risk of postpartum depression, empowers mothers, and involves fathers in the care of the newborn [5, 9].

Cultural norms influence perceptions of KMC and may affect the success of its adoption. On the personal level for the parent, the main issues with KMC are usually lack of time, lack of social support and medical care, as well as lack of family acceptance of this method of care [10]. Within the health system, the barriers are often financing, organization, and service delivery [11]. Other challenges in the implementation of KMC globally are differences in the recommended frequency and duration of skin-to-skin contact as well as the specific criteria for interrupting skin-to-skin contact. A lack of money to return for follow-up in the hospital, a lack of privacy in the hospital when performing KMC, caregivers' adherence to traditional newborn practices, and the stigma of having a preterm baby are other barriers described in the literature [10, 11].

There is little research on KMC in community-based settings. A study from rural Bangladesh investigating survival in community-based KMC suggests that more research is needed to understand the effect of community-based KMC on neonatal mortality and whether KMC

is optimally implemented [12]. Another study of KMC in community settings in Bangladesh found that the workload at home was a barrier to mothers performing KMC, and that a supportive environment both at home and in the community was important [13].

Bangladesh, a lower-middle-income country, suffers from one of the highest neonatal mortality rates in the word. According to the last Bangladesh Demographic and Health Survey conducted with a nationally representative sample in 2019, the mortality rate is 30 per 1000 live births. At the 67th World Health Assembly the government of Bangladesh endorsed the global Every Newborn Action Plan [14] and declared its commitment to reaching the sustainable development goal of reducing the neonatal mortality rate. As part of the strategy to address preventable neonatal deaths, KMC was integrated in care at facility level with continuation at home [15] for LBW and preterm babies. In 2016 KMC became one of the essential services to be provided to preterm and LBW babies. However, data on the percentage of LBW babies enrolled in facility-based KMC is not available, apart from the separate KMC Monthly Report Forms from the facilities providing KMC.

KMC is a complex intervention and the implementation of KMC is connected to the functioning of the health system and to cultural practices. Specific enablers and barriers must be identified in the local setting in order to facilitate the implementation of a KMC program [10]. KMC requires the active participation of health service providers, caregivers, and their family members. In addition, it requires cultural adaptation and behavioral interventions for appropriate adoption. Since KMC is a relatively new method of care in Bangladesh, it is time to systematically explore and document the areas for improvement and identify potential solutions for improving quality and subsequent scaling up of the program nationally.

The objective of this study was to explore caregivers' experiences of performing KMC in hospital settings and at home in Bangladesh, with the intention of identifying enablers and barriers to optimal implementation.

## Material and methods

A descriptive qualitative methodology was adopted. Semi-structured interviews were conducted with caregivers to newborns receiving KMC in the hospital or who had received KMC in the hospital and then been discharged one week prior to the interview date [16]. This study is reported in accordance with the Consolidated Criteria for Reporting Qualitative Research [17] (See S1 Appendix).

### Setting

The study was conducted in two hospitals in Dhaka, Bangladesh. The hospitals were selected based on their overall KMC utilization as well as in consultation with the National Newborn Health Program and Directorate General of Health Services. Convenience sampling was used. Hospital A in Mohammadpur Fertility Services and Training Centre is a secondary-level public hospital with 100 beds, of which six were dedicated to newborns. Three of these six beds were dedicated to KMC. Approximately five newborns received KMC there every month. Hospital B in the Institute of Child and Mother Health is a tertiary-level public hospital with 200 beds. Fifty-five beds were dedicated to newborns, with fifteen dedicated to KMC. Approximately fifty newborns received KMC there every month. Caregivers were able to perform KMC at any time in both hospitals.

In Hospital A there was a KMC corner with three beds, separated by curtains. Female and male visitors were allowed in the ward during the day. The staff were responsible for the KMC corner as well as the pediatric ward located in the same area. In Hospital B there was a KMC ward consisting of a large room without curtains separating the beds. The ward was staffed by

nurses on a rotation schedule. No male visitors were allowed, but the caregivers could have female relatives as visitors during the day. In hospital B there was a neonatal intensive care unit (NICU) outside the KMC ward, but the caregivers were only allowed there when the baby was to be breastfed. One bed was allocated to each caregiver in both hospitals.

In Hospital A there was no cost for KMC. Hospital B charged a fee for use of KMC equipment as well as a daily hospital rate. Both hospitals had a follow-up system with weekly follow-up until the baby was 40 weeks of corrected gestational age or had reached 2500 g. The caregivers were contacted by phone if they did not come to the follow-up.

## Participants

The study participants were caregivers who had experience of performing KMC. To be included in the study, a person had to be a caregiver currently performing KMC in the ward, or one who had been discharged one week earlier from the hospital after having performed hospital-based KMC. Convenience sampling was used to identify caregivers willing to participate in the interview during the data collection period. Consent was required to be included in the study. All caregivers who were asked agreed to participate in the study.

## Data collection

The interviews were conducted in August 2019 and March 2020 by the authors JS and HO with two different data collection teams. The first data collection focused on caregivers' experiences of performing KMC in the hospital. The second data collection focused on caregivers' experiences of having performed KMC in the hospital and then coming back for the one-week follow-up visit.

In total fifteen semi-structured interviews were conducted; ten in-ward interviews and five one-week follow-up interviews. Six in-ward interviews were conducted in Hospital A and four in Hospital B. The five follow-up interviews were all with participants from Hospital B.

The interview guide with semi-structured questions were developed by the first two authors with input from researchers from the International Centre of Diarrheal Disease Research, Bangladesh (icddr,b) and the Department of Women's and Children's Health, Uppsala University. Interview guide one was developed first, and the second interview guide was a modified version of the first one, with the same questions regarding the experience of performing KMC. The interview guides were developed in English and then translated into Bangla, the local language. The interview guides were pilot tested before finalization. The pilot test interview in the first data collection period was not included in the study. The pilot interview took place in Hospital A in the same settings as the other in-ward interviews.

Different aspects of the experience of performing KMC were explored in the interviews. The focus was on barriers to and enablers of skin-to-skin contact with the preterm/LBW baby and feeding practices. In the follow-up interviews the continuation of KMC at home after discharge was also explored.

Data collection one (August 2019), in-ward interviews 1–10: The interviews were conducted by a team consisting of the author JS the first author (a female pediatric nurse with an MSc in public health and previous training in qualitative methods and experience in conducting interviews), a male anthropologist with past experience of conducting interviews, and a female researcher from icddr,b with relevant experience. The interviews were conducted in the KMC wards in the participating facilities, while the baby was in the skin-to-skin position with the caregiver. Other caregivers were present in the room, but there was sufficient distance between them and the interviewee to ensure that the interview was not overheard. Interview

guide one (see S2 Appendix) was used for the in-ward interviews. The duration of the interviews was between 16 and 41 minutes, with an average of 31 minutes.

Data collection two (March 2020), follow-up interviews 11–15: The interviews were conducted by a team consisting of the author HO a female medical student with training in qualitative methods and interview methodology, and a female anthropologist from icddr,b with experience of conducting interviews. The interviews were conducted in a separate room in the hospital when the caregiver returned to the facility for the one-week follow-up or in the home of the caregiver one week after discharge from the hospital. Interview guide two (see S3 Appendix) was used for the follow-up interviews. The duration of the interviews was between 33 and 47 minutes, with an average of 36 minutes.

The data collection teams made it possible to conduct the interviews in the native language of the participants. The authors JS and HO were present during the interviews to answer questions arising during the interviews and to get a sense of the interview environment in order to be able to consider this when interpreting the results. Field notes were taken and were used to capture details of the interviews, including non-verbal communication. The results were discussed within the data collection team after each interview. When necessary, questions were adjusted for the next interview.

Informed, written consent was obtained from each participant before the interviews commenced (see S4 and S5 Appendices for consent forms for in-ward interviews and follow-up interviews). The interviews began by introducing the objectives of the study and clarifying that participation was voluntary and that the participant had the right to withdraw their consent at any time during the interview. The author present for the interview was then introduced to the participant and their interest in the research explained.

The interviews were conducted face to face in Bangla and were audio recorded. One in-ward interview was repeated due to the pain of the mother who was being interviewed on her first day of performing KMC. Recruitment for the in-ward interviews stopped when the team agreed that the data set was saturated. Saturation was considered obtained when only limited new information arose in the interviews. For the follow-up interviews, data collection stopped earlier than planned due to the onset of the Covid-19 pandemic. After preliminary analysis and discussion, the research team decided that saturation was considered to be achieved for the follow-up interviews.

All audio records were transcribed verbatim and translated into English. A selection of the interviews was translated back into Bangla by the author SMR, an experienced senior researcher, to ensure accuracy before coding.

## Data analysis

Thematic analysis with an inductive approach was used to analyze the interviews. Thematic analysis, as described by Braun and Clarke [18], is used to find patterns or themes within the data. The interviews were first read several times by JS and HO in order to gain an understanding of the data through paying attention to similarities and patterns. Thereafter, the data set was coded by JS and HO, part of it was coded together and part of it separately. Through an iterative process, subthemes and themes were identified from the data. The authors met several times and discussed the findings and agreed on the subthemes and themes. Word and Excel spreadsheets were used in the process of sorting the data. Participants did not provide feedback on the findings. To ensure trustworthiness, a checklist for criteria for thematic analysis was used during the process of analyzing the results [18]. Conformability was obtained by using quotations from the interviews to illustrate the informants' voices in the result. To ensure credibility the interview guides were similarly structured and covered the same areas.

Dependability was obtained by the work process described and a detailed description of the procedure as set out in the method section of this paper.

## Ethical considerations

Ethical approval was obtained from the Ethical Review Board at icddr,b (PR-19090 and PR-20013). The study was performed in accordance with the ethical standards in the Declaration of Helsinki [19]. Participants were informed about the objective of the study and told that participation was voluntary. Informed written consent was obtained from the participants before their inclusion in the study.

## Results

Of the fifteen caregivers interviewed, thirteen were mothers, one was a father, and one was a mother's sister. Background information about the caregivers was collected at the time when they consented to being enrolled in the study. The caregivers were between 18 and 39 years old and had been in the KMC ward/KMC corner between 2 and 8 days, with an average of 4 days for the in-ward interviews, and 2 to 9 days for the follow-up interviews, with an average of 6 days. Eight out of twelve mothers had been delivered through cesarean section. Data on mode of delivery are missing for the other four participants.

Analysis of the interviews led to the identification of three themes: *conducive conditions*, *an empowering process*, and *suboptimal implementation* and ten subthemes, as shown in Table 1. To illustrate the informants' voices, quotations are presented in the results.

### Conducive conditions

The conditions were conducive to performing KMC. The caregivers displayed acceptance of this method of care both in hospital and at home. The healthcare staff were perceived as supportive and the physical environment as facilitating. The social network was essential for the caregiver to perform KMC both in hospital and home.

**Displaying acceptance.** KMC was a new concept for all the caregivers. However, they expressed positive feelings about KMC and liked the idea as they learned more about it. Almost all of them expressed satisfaction with their hospital stay and displayed trust in the competence of the staff by following the doctors' and nurses' instructions on how to perform KMC; for example, by giving medicines as prescribed and expressing willingness to come to the hospital for follow-up. At both in-ward interviews and follow-up interviews, the caregivers said that KMC was feasible at home. *"Actually, I thought about what they suggested, and it seems they*

**Table 1. Themes and subthemes representing caregivers' experiences of KMC.**

| Themes | Subthemes |
|---|---|
| *Conducive conditions* | *Displaying acceptance* |
| | *Supportive healthcare staff* |
| | *Facilitating structures* |
| | *Importance of social networks* |
| *An empowering process* | *Becoming motivated* |
| | *Strengthening interaction with baby* |
| | *Instilling agency* |
| *Suboptimal implementation* | *Struggling to succeed* |
| | *Disregard for mothers' needs* |
| | *Delayed initiation* |

*have suggested good ideas to me. I did not get this treatment before when my other two children were born. I did not even know about it, after being admitted to this hospital I got to know about it. It felt good."* (Interview 1)

**Supportive healthcare staff.** Both doctors and nurses were appreciated as teachers of KMC as they instructed the caregivers on the method, as well as encouraged and supported them in the performance of KMC. Caregivers were informed that the aim was to keep the baby in the KMC position for a minimum of 20 hours a day and were instructed on how to position the baby. They were also informed about the benefits for the baby of skin-to-skin contact, including weight gain, brain and lung development, and a stronger immune system. The nurses had an important facilitating role in the KMC ward. They came in regularly and helped the mothers to tie and untie the binder and encouraged the caregivers. *"I have been shown all these today by someone. She asked me to show her how I am carrying the baby, I showed and then she told me that I am good at it."* (Interview 3)

**Facilitating structures.** Caregivers stated that the pedagogical pictures, informative TV screens, and the provision of the KMC binder favored the initiation of KMC. Everything needed was available in the KMC ward, including running water, soap, a sink, and diapers. They appreciated that the protected time in the hospital, with no housework, allowed them to focus attention on their baby. *"There were no household troubles, no extra troubles, and just younger baby at hospital."* (Interview 5)

The fact that only women were allowed in the KMC ward made it easier for the mothers to breastfeed the baby. Important routines mentioned were daily updates on the baby's weight and the frequent measurement of temperature, which gave understandable feedback on the baby's health to the caregivers. "*They measure weight every day, and it (the baby) is growing every day."* (Interview 3)

**Importance of social networks.** The caregivers stated that family support was essential, both at home and in the hospital. Without such support, it would have been difficult to perform KMC. A female member of the family was often present in the hospital to assist the mother with the daily routines and sometimes took turns providing skin-to-skin care. During the first period at home, the presence of relatives relieved the caregiver of household work. At home, caring for the newborn was a responsibility shared with the whole family. The husbands' support was appreciated. In the hospital his support consisted mainly of paying expenses and buying and bringing medicines to the baby and the mother. After coming home, the husbands were described as taking on responsibility for childcare. *"It was difficult for me to stay alone in the KMC ward. I had a little trouble when my sister left."* (Interview 11)

## An empowering process

The caregivers' experienced KMC as empowering. They became motivated by seeing their babies benefit from the care, and interaction between the caregiver and the newborn was strengthened. The process instilled agency in the caregivers, making them advocates for this method of care.

**Becoming motivated.** The caregivers said that they understood the connection between KMC and a healthy baby. Their desire to see their baby growing and staying healthy motivated them to perform KMC. Health aspects that the caregivers emphasized were the positive effects on brain development, the immune system, breastfeeding, and weight gain.

As the treatment proceeded, the caregivers paid attention to the progress of the baby. They noticed the benefits of KMC such as weight gain, increased appetite and improved breastfeeding and better sleep, as well as how the baby was kept warm through the skin-to-skin contact.

*"I understood that this will heal my babies, they will become healthier. The motherhood bond will become stronger, that is why I got much more interested."* (Interview 8)

**Strengthening interaction with baby.** Performing KMC was an opportunity for bonding. The time spent together in the KMC ward enabled the caregiver and the baby to get to know each other. The caregivers learned their newborn baby's signals and were attentive to them when they were crying, sleeping, and eating, and hygiene was an essential aspect when caring for the baby. The caregivers pointed out how KMC benefited both themselves and their baby, and they expressed happiness at being close to the baby. *"Keeping it (the baby) close to my chest means that there is another kind of attraction growing."* (Interview 14)

**Instilling agency.** Performing KMC and seeing the positive outcomes for the baby and for themselves gave caregivers a sense of empowerment. They saw no obstacles to doing light housework or walking and talking while the baby is skin-to-skin. A mother said that it was not a problem having the baby tied to her chest while sleeping. A father of twins said that he felt lucky to get the opportunity to be close to the babies and provide skin-to-skin care. The caregivers were confident that people at home would accept their having the baby skin-to-skin. Most of them were willing to come for follow-up. *"It is usually expensive keeping babies in the incubator. On top of that you cannot take care of your babies, you cannot see them. Now my babies are on my chest, I feel confident, and it is saving my money too."* (Interview 9)

Having had a good experience of KMC, several of the caregivers had become advocates of the method. They now felt confident and willing to tell and teach others (husband, family members or neighbors) about KMC. *"Yes, because people will ask me why I am keeping her like this. So, I will say that the doctor suggested it. And will add the reasons behind why I am holding her this way that it will counter her malnutrition and bring her quick recovery. I will tell them that they too can follow the same procedure if any infants need nutrition and care."* (Interview 6)

## Suboptimal implementation

The results also revealed barriers to optimal KMC practice. The caregivers had difficulty in keeping the baby in KMC-position. Mothers who had undergone a cesarean section struggled with pain, which hampered skin-to-skin performance. A baby's condition had often deteriorated before it was admitted to the KMC ward, which together with the separation of caregiver and baby in the NICU and after cesarean section led to delayed initiation of KMC.

**Struggling to succeed.** In the hospital, according to the caregivers, it was more common to hold the baby for two hours at a time, and there were many disruptions during the day. The caregivers struggled with the position of the baby and were bothered when the baby was crying, which made them open the binder. They spoke of the need for assistance in tying and untying the baby, and said that they felt insecure about managing this on their own. When going to the bathroom, eating, or sleeping at night, the baby was often left on the bed. Only a few caregivers stated they were sleeping with the baby tied to their chest. At home, in general, KMC was provided less than in the ward and for shorter sessions. *"One day the maximum was half an hour, then I take the baby off. After this I am trying for half an hour like this again, but I could not keep it that way continuously. Because he was crying."* (Interview 12)

It was difficult to perform KMC and housework at the same time. With other children that needed attention as well as light chores to be done by the mother, the baby was left alone on the bed when sleeping. Caregivers said that on the one hand they were worried about how to manage caring for the baby and housework when they returned home, and on the other hand they felt that they needed to go home to take care of the household and the rest of the family. Having twins who both needed KMC was an extra challenge as regards the duration of KMC for each baby. There was also a struggle for space, with only one hospital bed, as well as an

increased need to have an extra family member as support. *"I can eat keeping them (twins) like this. But other heavy chores like dishwashing, cooking, these are tough."* (Interview 1)

Some beliefs also led to disruptions of the skin-to-skin contact. A common idea was that the baby cried because it was too warm, or that the baby cried because it was unable to move its hands and feet, and wanted freedom.

Out-of-pocket-expenses were an issue for caregivers staying in the KMC ward. Even if they did not need to pay for the bed in one of the study hospitals, money was needed to buy medicine for the mother and to pay the expenses for transporting relatives visiting the caregiver. *"We came here from the village, and incurred a lot of transportation costs,"* (Interview 9)

**Disregard for mothers' needs.** Some parts of the hospital environment were identified as uncomfortable, and the caregivers wished for more privacy in the ward. They felt uneasy breastfeeding when other people were around and when male figures opened the door to the ward. So they often covered themselves with scarves when breastfeeding. Sometimes the caregivers felt too hot in the hospital, wished for access to proper air and sunlight, and longed to go home. *"Yes, I feel secure. But, at times many people come to visit here. At that time it feels a bit uneasy to feed in front of that crowd."* (Interview 6)

There was only one nurse on KMC duty at a time, and that nurse was not always present in the ward to support the caregivers. Some caregivers wished for more presence and support from the hospital staff. Mothers who had delivered through cesarean section were often in pain, which hampered the performance of skin-to-skin. *"My main problem is that because I went through cesarean section, I have stitches around my stomach, and if I hold the baby, it hurts a lot."* (Interview 1)

**Delayed initiation.** Deteriorated conditions of the baby were often needed for admission to the KMC ward and most of the time, several days passed between delivery and commencement of skin-to-skin contact. The health issues could be jaundice, convulsions, brain infections, or a failure to gain weight. *"They told us it would have been better if we were admitted here long ago, but no doctor ever said anything like this before."* (Interview 8)

Various situations led to the caregiver and baby being separated in the first days after delivery, delaying the start of skin-to-skin contact. First, if the baby was admitted to the NICU, the mother and baby were separated, and the mother was only called in to see the baby when the baby needed to be breastfed. Second, when the mother had undergone a cesarean section, she was separated from her baby for a few days before meeting again in the KMC ward.

## Discussion

The objective of this study was to explore caregivers' experience of performing KMC in Bangladesh, both in hospital settings and during continuation at home, with the intention of identifying enablers and barriers to optimal implementation.

The results showed that the conditions were conducive to KMC and that carrying out KMC was an empowering process for the caregiver. In the literature, KMC is often described as difficult and strenuous, and it is said that healthcare staff struggle to get parents to practice it. Our results indicate that this might not be the full picture. However, the results also reveal barriers to optimal KMC practice with suboptimal implementation and a lack of stringency. The challenges are complex and may need a multitude of interventions in order to support the caregivers in performing skin-to-skin contact for longer periods. Strengthening the involvement of other family members may be a feasible way to do this. We suggest this be further explored with involvement of the caregivers.

## Conducive conditions

The caregivers in this study had positive attitudes to KMC and were open to adopting the practice after being introduced to it, for it was new knowledge to them. This is also the case in similar settings in Malawi, where KMC was seen as feasible and mothers showed acceptance after becoming aware of its benefits [20, 21]. Lack of knowledge has been described as a barrier to optimal KMC practice [22–24]. The healthcare staff were important teachers and facilitators for the caregivers staying in the hospital with the baby. Both doctors and nurses had key roles as informants introducing KMC to the caregivers and their relatives. Education on the health benefits of the practice and feedback to the caregivers on the development of the baby, such as weight gain, facilitated KMC in our study and in previous studies [11, 20, 22, 23]. Supportive healthcare staff increase parental readiness and are important facilitators for KMC [11, 25]. Therefore, the importance of their role needs to be highlighted and their presence in the ward facilitated in order to strengthen KMC practices.

Family support was essential for the caregivers to perform KMC both in-ward and at home by helping with daily tasks, dealing with housework, and co-providing KMC [13, 20]. The involvement of the father has been shown to have a large impact on KMC uptake, as well as the involvement of grandmothers as co-providers [11, 26]. Including them in both education and practical performance has the potential to support KMC practice. Increasing fathers' involvement in KMC might, however, be challenging due to cultural habits and social structures [27].

Increasing awareness of the importance of follow-up is highlighted in the MaMoni HSS Report [28] that evaluated KMC in Bangladesh. The caregivers in our study were positive and willing to come for follow-up after discharge, as has also been the case in similar settings [29]. Because the distance from the hospital hinders people from coming for follow-up, finding alternative solutions for follow-up could be helpful in optimizing this component of KMC. A study by Ehtesham Kabir et al. [30] found that the follow-up rate in Bangladesh was low. The first follow-up had the highest attendance, but attendance decreased thereafter.

## An empowering process

The time in the KMC ward was an opportunity for bonding with the baby and instilled agency in the caregivers. The ward was a shared space with other caregivers caring for their baby skin-to-skin, which created an opportunity for peer support, motivating them to perform KMC as they learned from each other. These findings are also in line with the literature [31, 32]. Learning about KMC and having the time to care for their baby strengthened the caregivers and made them confident in their caregiving capacity. This experience of performing KMC was so empowering that they said they would spread knowledge about this method of care in their community. Increased knowledge and acceptance in the society can support the uptake of KMC.

## Suboptimal implementation

We saw missed opportunities for babies to access the benefits of KMC. The babies who received KMC in our study all had other medical issues besides being preterm or LBW. Babies who were in an unstable condition and being treated in the NICU were separated from the parent, leading to delayed initiation of KMC and stress among the caregivers. This issue has been highlighted in similar settings [24]. Having the baby close by made the caregiver feel good. It was perceived as a contrasting and positive experience compared to having the baby admitted to the NICU, where the caregivers were not allowed to stay. We found that separation between the caregiver and baby should be prevented in order to achieve the benefits of early

initiation of KMC [5, 7–9]. The importance of not separating the mother and baby is also emphasized by Nyqvist et al. [33] and by WHO study group [34].

The fact that most of the babies in our study had aggravating conditions as well as being preterm/ low birthweight when admitted for KMC may be a sign that too few low birthweight babies were being admitted to KMC. A rapid assessment of the implementation of KMC in different countries including Bangladesh found that only 9% of babies who could benefit from KMC were admitted to it [35]. Ehtesham Kabir et al. [30] found that the low proportion of eligible babies enrolled in KMC in Bangladesh is one of the challenges. The low uptake to KMC needs to be addressed in order to see its benefits on a large scale.

The mothers had difficulties in keeping the baby skin-to-skin and breastfeeding. We found the main reasons were mental challenges when the baby was crying or was perceived to be showing signs of discomfort and physical challenges related to pain after a cesarean section. The lack of privacy also made the mothers uncomfortable when breastfeeding, as was noted by Smith et al. [11].These challenges contribute to difficulty in meeting the 20 hours per day of skin-to-skin contact recommended in the National Guidelines [36].

The nurse assigned to the KMC unit could not always be present in the ward, and the caregivers expressed a wish for more support from the staff to help them in tying and untying the KMC binder. The presence of healthcare staff in the ward could be strengthened to meet this need. Continuous support and designated health personnel for skin-to-skin care have the potential to improve KMC practice [11, 37]. Evidence indicates a connection between duration of skin-to-skin care and early initiation of breastfeeding [38–40]. Since the recommendations and the reality when it comes to duration of skin-to-skin are quite different, personnel in the wards should be reminded that every hour of skin-to-skin contact contributes to the health of the babies. This is something that can also be highlighted to motivate caregivers to perform skin-to-skin contact and the healthcare staff to support longer duration of skin-to-skin sessions.

Our study noted that out-of-pocket costs were an issue for people to stay in the KMC ward. One of the study hospitals provided KMC care free, while the other hospital charged a fee for KMC care. This may have affected whether people could afford to stay, as well as the length of the stay. The consequence of this were not investigated in this study but costs have been described as a barrier by Yue [31]. Many of the caregivers expressed the belief that it would be good if the care were given free.

## Strengths and limitations

There were no follow-up interviews for hospital A due to the Covid-19 situation, and further data collection for the follow-up interviews was not possible due to the sudden lockdown in the country. However, after analyzing the in-ward interviews and the follow-up interviews together, saturation was seen as fulfilled for both the in-ward interviews and the follow-up interviews.

Another limitation could be that the in-ward interviews were conducted in the KMC ward/ KMC corner and other caregivers were present, which could have prevented the caregivers from expressing their experiences freely. The babies in the ward needed neonatal care and were in skin-to-skin position when the interviews were conducted. It had been decided, in consultation with the healthcare staff, that the caregiver should not leave the ward and that the caregiver and the baby should not be separated, and thus the interviews were conducted in the ward. In order to promote a safe environment for the interviews, the nurse in the ward and the healthcare staff were not present in the room at the time of the interview. The beds were also far enough apart to prevent others from overhearing the conversation.

The in-ward interviews were conducted while the caregivers performed KMC during their stay in the hospital. The caregivers had varying number of days of experience of performing KMC, which may have affected their responses. The fact that there were two or three people in the room together with the caregiver during the interview may also be seen as contributing to a power imbalance that may have affected caregiver's to say what they thought the research team wanted to hear.

Another limitation is that the authors JS and HO did not speak Bangla, the language used for the interviews. However, as described in the method section, the results were discussed by the data collection team after each interview. When needed, questions were adjusted before the next interview based on clarification and discussion with the first authors about what had come up in the interview. The role of JS and HO, who were present during the interviews, was to supervise and to understand the interview environment. After transcription, the interviews were translated into English. There is a risk that some information may have been lost or wrongly interpreted. This risk was reduced by selecting some of the interviews and translating them back into Bangla so that the interview could be checked by an experienced researcher to ensure the accuracy of the translation before coding.

A strength of the study is that a reflexive approach was adopted. There was close collaboration within the data collection teams as well as among the authors when analyzing and presenting the data. Another strength of the study is the use of semi-structured interviews to capture the perspective of the caregivers both during their time in the KMC ward and at follow-up. Another strength is the focus on both enablers and barriers to optimal KMC implementation. The hospitals included in the study provide care to a broad patient group including different socioeconomic groups. One of the hospitals has a broad uptake of patients including those living outside Dhaka.

## Implications for clinical practice and future research

The study identified a number of measures that have potential to increase the uptake of KMC. These include addressing the barriers identified in this study as well as strengthening and making more use of facilitating factors such as the socially and culturally conducive conditions and the positive experiences of the caregivers.

- The caregivers' acceptance of KMC suggests that the idea of KMC should be introduced in antenatal care for mothers in order to increase awareness about the method before they are admitted.

- The empowerment experienced by the caregivers should be harnessed to spread knowledge about KMC in the community and provide opportunities for them to act as role models in the ward.

- More use should be made of social networks by facilitating the presence of family members in the KMC ward in order to help the caregiver provide skin-to-skin contact.

- Mothers' motivation should be increased by providing feedback on the baby's condition while in-ward and in follow-up visits.

- High presence of healthcare staff is needed for continuous support to overcome the difficulties of keeping the baby skin-to-skin for long periods.

- Steps should be taken to improve the possibility of privacy in the wards.

- Mothers' need for adequate pain relief while admitted to the hospital should be addressed, including educating the staff about its importance.

- The causes of late initiation and short duration of KMC should be investigated to determine potential solutions.

- Options for preventing separation of caregiver and baby in the NICU should also be explored.

This study adds new knowledge about enablers and barriers to KMC in hospital settings and to continuation of KMC at home in Bangladesh. The result has clinical implications that might be useful to consider when scaling up KMC.

## Conclusion

The social and cultural conditions for the caregivers to perform KMC as well the empowerment the parents felt in their roles as caregivers when performing KMC are facilitating factors. Initial separation and late initiation of KMC, and disregard for mothers' need for care and support were barriers to optimal practice, leading to missed opportunities. These facilitators and barriers need to be addressed in order to succeed with scaling up the national KMC program in Bangladesh.

## Supporting information

**S1 Appendix. Consolidated checklist.**
(DOCX)

**S2 Appendix. Interview guide for in-ward interviews.**
(DOCX)

**S3 Appendix. Interview guide for follow-up interviews.**
(DOCX)

**S4 Appendix. Consent form for in-ward interviews.**
(DOCX)

**S5 Appendix. Consent form for follow-up interviews.**
(DOC)

## Acknowledgments

We would like to thank all the caregivers participating in the study for sharing their experience. We also thank physicians, nurses, and the managers Dr. Md. Munisuzzaman Siddiqui, Director, and Dr. Md. Mozibur Rahman, Associate Professor of Neonatalogy, for facilitating the data collection at the Mohammadpur Fertility Services and Training Centre (MFSTC) and the Institute of Child and Mother Health (ICMH), Dhaka, Bangladesh.

## Author Contributions

**Conceptualization:** Johanna Sjömar, Hedda Ottesen, Goutum Banik, Ahmed Ehsanur Rahman, Ylva Thernström Blomqvist, Syed Moshfiqur Rahman, Mats Målqvist.

**Data curation:** Johanna Sjömar, Hedda Ottesen, Goutum Banik, Ahmed Ehsanur Rahman, Syed Moshfiqur Rahman, Mats Målqvist.

**Formal analysis:** Johanna Sjömar, Hedda Ottesen, Mats Målqvist.

**Funding acquisition:** Johanna Sjömar, Hedda Ottesen, Ylva Thernström Blomqvist.

**Investigation:** Johanna Sjömar, Hedda Ottesen, Goutum Banik, Ahmed Ehsanur Rahman, Syed Moshfiqur Rahman, Mats Målqvist.

**Methodology:** Johanna Sjömar, Hedda Ottesen, Goutum Banik, Ahmed Ehsanur Rahman, Ylva Thernström Blomqvist, Syed Moshfiqur Rahman, Mats Målqvist.

**Project administration:** Johanna Sjömar, Hedda Ottesen, Goutum Banik, Ahmed Ehsanur Rahman, Ylva Thernström Blomqvist, Syed Moshfiqur Rahman, Mats Målqvist.

**Resources:** Johanna Sjömar, Hedda Ottesen, Ylva Thernström Blomqvist, Mats Målqvist.

**Software:** Johanna Sjömar, Hedda Ottesen, Mats Målqvist.

**Supervision:** Ylva Thernström Blomqvist, Syed Moshfiqur Rahman, Mats Målqvist.

**Validation:** Johanna Sjömar, Hedda Ottesen, Ylva Thernström Blomqvist, Syed Moshfiqur Rahman, Mats Målqvist.

**Visualization:** Johanna Sjömar, Hedda Ottesen, Mats Målqvist.

**Writing – original draft:** Johanna Sjömar, Hedda Ottesen, Goutum Banik, Ahmed Ehsanur Rahman, Ylva Thernström Blomqvist, Syed Moshfiqur Rahman, Mats Målqvist.

**Writing – review & editing:** Johanna Sjömar, Hedda Ottesen, Goutum Banik, Ahmed Ehsanur Rahman, Ylva Thernström Blomqvist, Syed Moshfiqur Rahman, Mats Målqvist.

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
