## [Editor Report · Decision Letter 0]

14 Apr 2022

PONE-D-22-08128Exploring caregivers’ experiences of Kangaroo Mother Care in Bangladesh– a qualitative studyPLOS ONE

Dear Dr. Sjömar,

Thank you for submitting your manuscript to PLOS ONE. After careful consideration, we feel that it has merit but does not fully meet PLOS ONE’s publication criteria as it currently stands. Therefore, we invite you to submit a revised version of the manuscript that addresses the points raised during the review process.

We look forward to receiving your revised manuscript.

Kind regards,

Halimatus Sakdiah Minhat, DrPH

Academic Editor

PLOS ONE

Journal Requirements:

3. Please amend your manuscript to include your abstract after the title page.

Additional Editor Comments:

Dear authors

I'm pleased to inform you that your manuscript has been deemed suitable for publication in PLOS ONE. However, the manuscript requires MINOR REVISIONS before it can be reconsidered for publication. Please note that it is imperative for you to revise the manuscript according to comments and guidelines. Please also highlight in the manuscript where changes are made.

1. Please improve the methodology section in the abstract

2. The discussion should provide subheadings related to the themes identified and should be discussed accordingly. The existing discussion seems s bit superficial and lack of critical thinking and this should be improved.

Thank you for submitting your article to PLOS One
---

## [Author Response · Author response to Decision Letter 0]

24 May 2022

Manuscript Number PONE D-22-08128 

Response to Rewiewers, PLOS ONE

The authors would like to thank the reviewers for their input and highly valuable comments for the article titled “Exploring caregivers´ experiences of Kangaroo Mother Care in Bangladesh- a qualitative study”. We believe that the review has resulted in an improved manuscript. Point-to-point answers follow below. 

Revised text in the new version of our manuscript is marked in red.

An unmarked version of our revised paper without tracked changes has been upload as a separate file labeled ”Manuscript”.

Journal Requirements

Respons:

1. Thank you, we have made sure the manuscript meets PLOS ONE´s style requirements, including those for file naming.

2. Regarding the minimal dataset underlying the result in the manuscript: All relevant data are within the manuscript and its Supporting Information files. Raw data (i.e. transcripts and translations) cannot be shared publicly due to assurances of confidentiality given to participants at the time of consenting. icddr,b is the local collaborator and ethics approval was obtained from the Ethical Review Committee at icddr,b in Dhaka, Bangladesh. Because of the statutory requirements, internal data policies and regulations existing in the collaborating bodies along with the over-arching General Data Protection Regulation (GDPR), the data must be stored in institutional repository (storage platforms) and cannot be made directly accessible without a review of the request for access to data. Therefore, the data can be accessed only upon formal request that details the purpose of such request. The request will then be processed by the Data Repository Committee (DRC) at icddr,b. Any such request should be directed to the principal investigator Dr. Ahmed Ehsanur Rahman (email: ehsanur@icddrb.org) or to the Mr. M A Salam Khan, IRB coordinator secretariat ( email: salamk@icddrb.org)

3. The manuscript has now the abstract after the title page

4. We have included captions for the Supporting Information files at the end of the manuscript and updated any in-text citations to match accordingly. The reference list has been reviewed to ensure that it is complete and correct.

Additional Editor Comments

Respons:

1. Thank you for the valuable comments. The methodology section in the abstract has been modified and now includes more detail.

2. The discussion has been provided by subheadings related to the themes identified and is discussed accordingly. The discussion has been extended, and changes have been made both in the discussion relating to results and methods. 

Reviewers´comments

There were no comments submitted as an attachment file.

---

## [Decision Letter · Decision Letter 1]

14 Aug 2022

PONE-D-22-08128R1Exploring caregivers’ experiences of Kangaroo Mother Care in Bangladesh– a qualitative studyPLOS ONE

Dear Dr. Sjömar,

Thank you for submitting your manuscript to PLOS ONE. After careful consideration, we feel that it has merit but does not fully meet PLOS ONE’s publication criteria as it currently stands. Therefore, we invite you to submit a revised version of the manuscript that addresses the points raised during the review process.

We look forward to receiving your revised manuscript.

Kind regards,

Tai-Heng Chen, M.D.

Academic Editor

PLOS ONE

Journal Requirements:

Reviewers' comments:

Reviewer's Responses to Questions

**Comments to the Author**

1. If the authors have adequately addressed your comments raised in a previous round of review and you feel that this manuscript is now acceptable for publication, you may indicate that here to bypass the “Comments to the Author” section, enter your conflict of interest statement in the “Confidential to Editor” section, and submit your "Accept" recommendation.

Reviewer #1: (No Response)

Reviewer #2: All comments have been addressed

Reviewer #3: All comments have been addressed

Reviewer #4: (No Response)

Reviewer #5: (No Response)

2. Is the manuscript technically sound, and do the data support the conclusions?

Reviewer #1: Yes

Reviewer #2: Yes

Reviewer #3: Yes

Reviewer #4: Yes

Reviewer #5: Partly

3. Has the statistical analysis been performed appropriately and rigorously? 

Reviewer #1: N/A

Reviewer #2: Yes

Reviewer #3: Yes

Reviewer #4: Yes

Reviewer #5: I Don't Know

4. Have the authors made all data underlying the findings in their manuscript fully available?

Reviewer #1: Yes

Reviewer #2: Yes

Reviewer #3: Yes

Reviewer #4: Yes

Reviewer #5: Yes

5. Is the manuscript presented in an intelligible fashion and written in standard English?

Reviewer #1: Yes

Reviewer #2: Yes

Reviewer #3: Yes

Reviewer #4: Yes

Reviewer #5: No

6. Review Comments to the Author

Reviewer #1: Exploring caregivers’ experiences of Kangaroo Mother Care in Bangladesh– a qualitative study

Thank you for providing this opportunity for me to review this manuscript. Please see my comments as follows:

Title: the term of “caregivers” reflects who provide care from neonates in hospitals and not mothers or fathers of low-birth weight neonate. I recommend that authors replace this with another word.

Abstract:

1. Please write keywords of the study after abstract.

Methods

1. Please write where did you conduct the interviews? In hospital or home?

2. Because some of interviews were coincident with COVID-19 pandemic, may authors considered to do some interviews online?

Results

1. Please provide at least one “quate” for each subtheme.

2. How many primary codes authors derived from interviews?

3. Please differentiate themes from subthemes using different fonts or “italic”.

4. In this section readers like to know which participants perceived KMC as conductive conditions or perceived it with obstacle? I am talking about type of delivery that is so important to adopt KMC. Mostly women who underwent cesarean section are not able to have KMC during first week after CS.

Reviewer #2: Hello

Dear manager

I read the article titled " Exploring caregivers’ experiences of Kangaroo Mother Care in Bangladesh– a qualitative study".

I recommended that this paper be accepted.

Reviewer #3: Reviewer reports

Title: Exploring caregivers’ experiences of Kangaroo Mother Care in Bangladesh– a qualitative study

Reviewer: Mesfin Abebe

Thank you for the opportunity to review this manuscript I read the manuscript in detail and the manuscript is well-structured and written in a good manner. However, I raised some minor comments and suggestions below. I'm hopeful the comments below can help fill these gaps and improve the quality of the paper.

Minor revision

The paper's overall language quality is a little low; there are numerous grammatical errors throughout the entire manuscript. Before the next submission, I recommend that the entire manuscript be revised by a native English speaker.

Abstract: The abstract accurately reflects the details presented in the manuscript's body. However, the use of abbreviations in the abstract section is not recommended, and I recommend revising it.

Background: your background information is somewhat good, but didn’t show a gap in your study.

Methods:

1. Clearly state the sampling procedure you used to choose the 15 participants.

2. You only interview 15 mothers in order to answer your research question. Do you have saturation data based solely on interview methods of data collection?

3. Why were only semi-structured interviews considered?

4. What were the admission requirements? Was admission made immediately after birth, or was a period of stabilization in a special newborn care unit?

5. How long did it take from birth to admission to kangaroo care?

Results

1. The result is generally well written and has good thematic contents

Discussion:

1. The discussion is generally well-organized and incorporates existing literature. There might be more consideration given to the next steps in the research or implementation

2. What do mean skin-to-skin sessions?

3. Clear the phrase “out-of-pocket money”

Reviewer #4: (No Response)

Reviewer #5: from the beginning the title is interesting and well written and structured manuscript.The data set support the conclusion. how ever the paper has some short comings in regards to abstract, introduction,some part of methodology and references. in general the text alignment must be corrected.the abstract must contain main part of methods like number of participants, data collection tolls and study period. it is better not to use abbreviation in the abstract part do not use abbreviation before stating the whole word ,the statistical analysis also roughly explained and it lacks detailed explanation.

from introduction part it is better to start from the background instead of stating the problem. secondly you have to state the problem from broader to focus and from global to local and you have to add the location relating to the figure of problem . from introduction page 5 there is long paragraph with out citation so make it paraphrase or add reference in between.you have to explain the implication of this study from study conducted in Bangladesh regarding kangaroo mother care. it is better to cite recent and relevant reference.

7. PLOS authors have the option to publish the peer review history of their article (what does this mean?). If published, this will include your full peer review and any attached files.

Reviewer #1: **Yes: **Parvin Abedi

Reviewer #2: No

Reviewer #3: **Yes: **Mesfin Abebe

Reviewer #4: **Yes: **Gossa Fetene Abebe

Reviewer #5: No

---

## [Author Response · Author response to Decision Letter 1]

27 Sep 2022

Response to Reviewers September 2022

PONE-D-22-08128R1

Exploring caregivers’ experiences of Kangaroo Mother Care in Bangladesh– a qualitative study

PLOS ONE

The authors would like to thank the reviewers for their input and highly valuable comments for the article titled “Exploring caregivers´experiences of Kangaroo Mother Care in Bangladesh- a qualitative study”. We believe that the review has resulted in an improved manuscript. Point-to-point answers follow below and in the document named Response to PLOS ONE.

An unmarked version of our revised paper without tracked changes has been uploaded as a separate file labeled ”Revised Manuscript” and as well with trach changes named “Revised manuscript with track changes”.

Journal Requirements:

Thanks for the comment. The referencelist is checked and is complete and correct.

Reviewers' comments:

Reviewer's Responses to Questions

Comments to the Author

1. If the authors have adequately addressed your comments raised in a previous round of review and you feel that this manuscript is now acceptable for publication, you may indicate that here to bypass the “Comments to the Author” section, enter your conflict of interest statement in the “Confidential to Editor” section, and submit your "Accept" recommendation.

Reviewer #1: (No Response)

Reviewer #2: All comments have been addressed

Reviewer #3: All comments have been addressed

Reviewer #4: (No Response)

Reviewer #5: (No Response)

2. Is the manuscript technically sound, and do the data support the conclusions?

Reviewer #1: Yes

Reviewer #2: Yes

Reviewer #3: Yes

Reviewer #4: Yes

Reviewer #5: Partly

3. Has the statistical analysis been performed appropriately and rigorously?

Reviewer #1: N/A

Reviewer #2: Yes

Reviewer #3: Yes

Reviewer #4: Yes

Reviewer #5: I Don't Know

4. Have the authors made all data underlying the findings in their manuscript fully available?

Reviewer #1: Yes

Reviewer #2: Yes

Reviewer #3: Yes

Reviewer #4: Yes

Reviewer #5: Yes

5. Is the manuscript presented in an intelligible fashion and written in standard English?

Reviewer #1: Yes

Reviewer #2: Yes

Reviewer #3: Yes

Reviewer #4: Yes

Reviewer #5: No

6. Review Comments to the Author

Reviewer #1: Exploring caregivers’ experiences of Kangaroo Mother Care in Bangladesh– a qualitative study

Thank you for providing this opportunity for me to review this manuscript. Please see my comments as follows:

Title: the term of “caregivers” reflects who provide care from neonates in hospitals and not mothers or fathers of low-birth weight neonate. I recommend that authors replace this with another word.

Thanks for the comment. We have looked at this and discussed with language proof reader, who think that ’Caregiver’ is an appropriate term to use. Hence we like to keep it.

Abstract:

1. Please write keywords of the study after abstract.

Thanks for the comment. Keywords has been added after the abstract. 

Methods

1. Please write where did you conduct the interviews? In hospital or home? 

Thank you for the comment. This has already been addressed under the section ”data collection”. The in-wards interviews were performed in the ward in the hospital and the follow up interviews either in a separate room in the hospital or at home. 

2. Because some of interviews were coincident with COVID-19 pandemic, may authors considered to do some interviews online? 

Thanks for the comment. Due to the sudden changes in the country and the lock down due to the corona situation we decided not to perform interviews online, since this there was no familiarity with online interviewing at the time and that it would alter the interaction and process between interviewer and interviewee too much. 

Results

1. Please provide at least one “quote” for each subtheme. 

Thanks for the comment. This is already presented under each subtheme in the manuscript.

2. How many primary codes authors derived from interviews? 

Thanks for the comment. 1424 primary codes were derived from the interviews.

3. Please differentiate themes from subthemes using different fonts or “italic”. 

Thanks for the comment, this is done with different fonts according to the guidelines of PLOS ONE.

4. In this section readers like to know which participants perceived KMC as conductive conditions or perceived it with obstacle? I am talking about type of delivery that is so important to adopt KMC. Mostly women who underwent cesarean section are not able to have KMC during first week after CS. 

Thanks for the comment. We don´t have enough data for this and this was not investigated in this study. We saw that CS was an obstacle for the mothers. However, there were other obstacles as well, as presented in the manuscript, but we have not connected the answers to if the mother with obstacle had a cesarian section or not in this qualitative study.

Reviewer #2: Hello

Dear manager

I read the article titled " Exploring caregivers’ experiences of Kangaroo Mother Care in Bangladesh– a qualitative study".

I recommended that this paper be accepted.

Reviewer #3: Reviewer reports

Title: Exploring caregivers’ experiences of Kangaroo Mother Care in Bangladesh– a qualitative study

Reviewer: Mesfin Abebe

Thank you for the opportunity to review this manuscript I read the manuscript in detail and the manuscript is well-structured and written in a good manner. However, I raised some minor comments and suggestions below. I'm hopeful the comments below can help fill these gaps and improve the quality of the paper.

Minor revision

The paper's overall language quality is a little low; there are numerous grammatical errors throughout the entire manuscript. Before the next submission, I recommend that the entire manuscript be revised by a native English speaker. 

Thanks for the comment. The manuscript has been sent to a professional language editor and changes have been done accordingly.

Abstract: The abstract accurately reflects the details presented in the manuscript's body. However, the use of abbreviations in the abstract section is not recommended, and I recommend revising it. Abbreviation will be removed.

Background: your background information is somewhat good, but didn’t show a gap in your study. ?

Thanks for the comment. We have tried to state this from the litterature

Methods:

1. Clearly state the sampling procedure you used to choose the 15 participants. 

Thanks for the comment. This has been clarified in the methodsection.

2. You only interview 15 mothers in order to answer your research question. Do you have saturation data based solely on interview methods of data collection? 

Thanks for the comment. Yes, saturation have been reached through the 15 interviews.

3. Why were only semi-structured interviews considered? 

Thanks for the comment. Semistructured interviews were considered the best method according to the research questions.

4. What were the admission requirements? Was admission made immediately after birth, or was a period of stabilization in a special newborn care unit? 

Thanks for the comment. This was different among the admitted children and data is missing. It is not written in detail in the manuscript.

5. How long did it take from birth to admission to kangaroo care? 

Thanks for the comment. Yes, it is true, data of some children are missing but we say in general that there were several days of delay before admission to KMC Care. 

Results

1. The result is generally well written and has good thematic contents

Discussion:

1. The discussion is generally well-organized and incorporates existing literature. There might be more consideration given to the next steps in the research or implementation. 

Thank you for this comment. We hope that our results will assist in the coming implementation of KMC. More research is needed to follow that process.

2. What do mean skin-to-skin sessions? 

Thanks for the comment. This is explaind in the manuscript and means periods of skin to skin care. We have looked at this and tried to use the same words in the manuscript. 

3. Clear the phrase “out-of-pocket money”. 

Thanks for the comments. This has been explained on page 20 raw 437-441. It refers to the expenses related to medicines for the mother and transportation costs for relatives travelling to visit in the hospital.

Reviewer #4: (No Response)

Reviewer #5: from the beginning the title is interesting and well written and structured manuscript.The data set support the conclusion. how ever the paper has some short comings in regards to abstract, introduction,some part of methodology and references. in general the text alignment must be corrected.the abstract must contain main part of methods like number of participants, data collection tolls and study period. it is better not to use abbreviation in the abstract part do not use abbreviation before stating the whole word ,the statistical analysis also roughly explained and it lacks detailed explanation. From introduction part it is better to start from the background instead of stating the problem. secondly you have to state the problem from broader to focus and from global to local and you have to add the location relating to the figure of problem . from introduction page 5 there is long paragraph with out citation so make it paraphrase or add reference in between.you have to explain the implication of this study from study conducted in Bangladesh regarding kangaroo mother care. it is better to cite recent and relevant reference. 

Thanks for the comments. The abtract has been updated, abbreviations are taken away. In the introduction part the reference has been updated ant the long paragraph addressed. The methodology is explained in more detail with number of participants, data collection tools and study period. The data analysis has been revised with a more detailed explanation.

7. PLOS authors have the option to publish the peer review history of their article (what does this mean?). If published, this will include your full peer review and any attached files.

Do you want your identity to be public for this peer review? For information about this choice, including consent withdrawal, please see our Privacy Policy.

Reviewer #1: Yes: Parvin Abedi

Reviewer #2: No

Reviewer #3: Yes: Mesfin Abebe

Reviewer #4: Yes: Gossa Fetene Abebe

Reviewer #5: No

---

## [Decision Letter · Decision Letter 2]

10 Nov 2022

PONE-D-22-08128R2Exploring caregivers’ experiences of Kangaroo Mother Care in Bangladesh– a descriptive qualitative studyPLOS ONE

Dear Dr. Sjömar,

Thank you for submitting your manuscript to PLOS ONE. After careful consideration, we feel that it has merit but does not fully meet PLOS ONE’s publication criteria as it currently stands. Therefore, we invite you to submit a revised version of the manuscript that addresses the points raised during the review process.

We look forward to receiving your revised manuscript.

Kind regards,

Tai-Heng Chen, M.D.

Academic Editor

PLOS ONE

Journal Requirements:

Reviewers' comments:

Reviewer's Responses to Questions

**Comments to the Author**

1. If the authors have adequately addressed your comments raised in a previous round of review and you feel that this manuscript is now acceptable for publication, you may indicate that here to bypass the “Comments to the Author” section, enter your conflict of interest statement in the “Confidential to Editor” section, and submit your "Accept" recommendation.

Reviewer #3: All comments have been addressed

Reviewer #4: (No Response)

2. Is the manuscript technically sound, and do the data support the conclusions?

Reviewer #3: Yes

Reviewer #4: Yes

3. Has the statistical analysis been performed appropriately and rigorously? 

Reviewer #3: Yes

Reviewer #4: Yes

4. Have the authors made all data underlying the findings in their manuscript fully available?

Reviewer #3: Yes

Reviewer #4: (No Response)

5. Is the manuscript presented in an intelligible fashion and written in standard English?

Reviewer #3: Yes

Reviewer #4: Yes

6. Review Comments to the Author

Reviewer #3: I read the manuscript again in detail and the manuscript is well-structured and written in a good manner. The author addressed all my comments and I recommended to accept the paper.

Reviewer #4: Manuscript Number: PONE-D-22-08128R1

Manuscript Title: Exploring caregivers’ experiences of Kangaroo Mother Care in Bangladesh– a qualitative study.

Thank you very much, Dr. Tai-Heng Chen, Academic editor of the PLOS ONE journal for giving me this opportunity to review this scholarly work which was grounded on a very interesting topic “Exploring caregivers’ experiences of Kangaroo Mother Care in Bangladesh– a qualitative study”.

I would also appreciate the authors of this manuscript for investigating this very important topic and coming up with such interesting findings. I have been impressed with the flow, organization, and way they synthesize the research from beginning to end. All the parts of the manuscript have been clearly stated and narrated. I appreciate the authors for their devoted work. Having this in mind, I do have some questions, comments, and suggestions to be addressed.

I had put all this concerns previously in the "Upload reviewer attachement section", but it was not adderessed.

General questions:

1. What type of qualitative study it was? Was it phenomenological, Ethnographic, Grounded theory, or ……

2. What were the questions of your research?

In the methods section:

In the data collection sub-section;

1. When, and where had the pilot study been done? Was it done in a similar setting or not?

2. Why the interview was not held in a separate room? Because, due to the fear of other people around the wards/interview areas, the participant may not give ample pieces of information which may hinder your findings. To elaborate more, there might be certain sensitive issues that might be positive or negative effects on the implementation of KMC. But, due to the fears of other caregivers other than the teams that participated in the data collection, the interviewed mothers/caregiver could be deterred from telling the truth, which may highly affect the result of your work. Please, consider this issue and amend it as much as possible.

3. When data saturation were determined? To clarify more, how many times the data were repeated to say the data were saturated? Please, make it clear!

In the strength and limitations:

1. One of the limitations of the study was, no follow-up study in hospital A, due to the outbreak of the COVID-19 pandemic. So, why not you consider other possible preventive strategies and continue the follow-up study? For example, by wearing a face mask you can continue the study as intended.

2. In general, what did you do to handle the impacts of the pandemic on your research findings? Please, incorporate as a strength of your study, if any activities were done to alleviate the impacts of the pandemic.

3. The other limitation of the study was, that there were two-three people in the room together with the caregiver during the time of the interview. So, why not you did the interview in a separate room or lonely? What were the challenges? Please, include the challenges, if any. If there were no challenges to doing the interview in a separate room or lonely, the way you did the interview was not scientifically acceptable.

7. PLOS authors have the option to publish the peer review history of their article (what does this mean?). If published, this will include your full peer review and any attached files.

Reviewer #3: **Yes: **Mesfin Abebe

Reviewer #4: **Yes: **Gossa Fetene Abebe

---

## [Author Response · Author response to Decision Letter 2]

18 Nov 2022

Response to Reviewers 

The authors would like to thank the reviewers for their input and highly valuable comments for the article titled “Exploring caregivers´experiences of Kangaroo Mother Care in Bangladesh- a qualitative study”. We believe that the review has resulted in an improved manuscript. Point-to-point answers follow below. 

Reviewer #3: I read the manuscript again in detail and the manuscript is well-structured and written in a good manner. The author addressed all my comments and I recommended to accept the paper.

Reviewer #4: Manuscript Number: PONE-D-22-08128R1

Manuscript Title: Exploring caregivers’ experiences of Kangaroo Mother Care in Bangladesh– a qualitative study.

Thank you very much, Dr. Tai-Heng Chen, Academic editor of the PLOS ONE journal for giving me this opportunity to review this scholarly work which was grounded on a very interesting topic “Exploring caregivers’ experiences of Kangaroo Mother Care in Bangladesh– a qualitative study”.

I would also appreciate the authors of this manuscript for investigating this very important topic and coming up with such interesting findings. I have been impressed with the flow, organization, and way they synthesize the research from beginning to end. All the parts of the manuscript have been clearly stated and narrated. I appreciate the authors for their devoted work. Having this in mind, I do have some questions, comments, and suggestions to be addressed.

I had put all this concerns previously in the "Upload reviewer attachement section", but it was not adderessed.

General questions:

1. What type of qualitative study it was? Was it phenomenological, Ethnographic, Grounded theory, or ……

Thanks for the comment. The study is an explorative descriptive qualitative study. Now the title is changed to Exploring caregivers´s experiences of Kangaroo Mother Care in Bangladesh- a descriptive, qualitative study.

2. What were the questions of your research?

Thanks for the comment. The question of the research is stated as the objective of the study in the manuscript. The objective of this study was to explore caregivers' experiences of performing KMC in hospital settings and after continuation at home in Bangladesh, with the intention to identify enablers and barriers for optimal implementation.

In the methods section:

In the data collection sub-section;

1. When, and where had the pilot study been done? Was it done in a similar setting or not?

Thanks for the comment. The pilotinterview was performed in one ot the 2 study hospitals, the day before the first interview for the datacollection. The interview was not included in the study as privacy was not ensured during this interview. This has been clarified in the manuscript.

2. Why the interview was not held in a separate room? Because, due to the fear of other people around the wards/interview areas, the participant may not give ample pieces of information which may hinder your findings. To elaborate more, there might be certain sensitive issues that might be positive or negative effects on the implementation of KMC. But, due to the fears of other caregivers other than the teams that participated in the data collection, the interviewed mothers/caregiver could be deterred from telling the truth, which may highly affect the result of your work. Please, consider this issue and amend it as much as possible.

Thanks for the important comment. The children at the ward were in need of neonatal care and was in skin to skin position when the interview were conducted. Together with the healthcare staff it was decided that the child should not leave the ward, which is why the interviews were conducted in the ward. In order to ensure a safe environment for the interviews the nurse in the ward the healthcare staff was not present in the room at the time of the interview. In that way, privacy was ensured despite being held in the ward. The distance between the beds were also large enough to prevent the other people in the ward to overhear the conversation. In the second set of interviews, conducted at the one-week follow-up, the interviews were held in a separate room with no other patients or healthcare staff present. This has been clarified now in the manuscript.

3. When data saturation were determined? To clarify more, how many times the data were repeated to say the data were saturated? Please, make it clear!

We noticed the answers were similar for the in- ward interviews until PI saw the themes were repeated. Follow up interviews were finished earlier but when analyzing the result form the follow-up interviews we saw similar themes and therefor these interviews was considered to be saturated. This has been updated in the manuscript.

In the strength and limitations:

1. One of the limitations of the study was, no follow-up study in hospital A, due to the outbreak of the COVID-19 pandemic. So, why not you consider other possible preventive strategies and continue the follow-up study? For example, by wearing a face mask you can continue the study as intended.

Thanks for the comment. The clinic decided the interviews could not be continued due to the risk of spread of the coronavirus and the lock down in the whole country. When the society was adopted to digital solutions it was not feasible to continue the follow up interviews.

2. In general, what did you do to handle the impacts of the pandemic on your research findings? Please, incorporate as a strength of your study, if any activities were done to alleviate the impacts of the pandemic.

Thanks for the comment. Due to the sudden lock down in the country due to Corona we couldn’t continue the data collection for the follow-up interviews. During the analysis we found we got enough of data and saturation were fulfilled. This has been discussed in the manuscript.

3. The other limitation of the study was, that there were two-three people in the room together with the caregiver during the time of the interview. So, why not you did the interview in a separate room or lonely? What were the challenges? Please, include the challenges, if any. If there were no challenges to doing the interview in a separate room or lonely, the way you did the interview was not scientifically acceptable.

Thanks for the comment. Regarding the first point raises, notes were taken on the environment during the interviews by the first two authors, they did not speak Bangla and they were available to clarify questions if needed, to come up with follow-up questions to fully understand the experience and opinions of the caregivers, as well as to take notes on the interview environment. So therefor it was not only one person performing the interview. 

Regarding the next point why the interview was not held in a separate room, please see the same response at point 2 under method section. The children at the ward were in need of neonatal care and supervision and was in skin to skin position when the interviews were conducted. Together with the healthcare staff it was decided that the child should not leave the ward, which is why the interviews were conducted in the ward ant not in a separate room. In order to ensure a safe environment for the interviews the healthcare staff was not present in the room at the time of the interview. In that way, privacy was ensured despite being held in the ward. The distance between the beds were also large enough to prevent the other people in the ward to overhear the conversation. In the second set of interviews, conducted at the one-week follow-up, the interviews were held in a separate room with no other patients or healthcare staff present. This has all been clarified in the manuscript.

---

## [Decision Letter · Decision Letter 3]

27 Dec 2022

Exploring caregivers’ experiences of Kangaroo Mother Care in Bangladesh– a descriptive qualitative study

PONE-D-22-08128R3

Dear Dr. Sjömar,

We’re pleased to inform you that your manuscript has been judged scientifically suitable for publication and will be formally accepted for publication once it meets all outstanding technical requirements.

Kind regards,

Tai-Heng Chen, M.D.

Academic Editor

PLOS ONE

Reviewers' comments:

Reviewer's Responses to Questions

**Comments to the Author**

1. If the authors have adequately addressed your comments raised in a previous round of review and you feel that this manuscript is now acceptable for publication, you may indicate that here to bypass the “Comments to the Author” section, enter your conflict of interest statement in the “Confidential to Editor” section, and submit your "Accept" recommendation.

Reviewer #3: All comments have been addressed

Reviewer #4: All comments have been addressed

2. Is the manuscript technically sound, and do the data support the conclusions?

Reviewer #3: Yes

Reviewer #4: Yes

3. Has the statistical analysis been performed appropriately and rigorously? 

Reviewer #3: Yes

Reviewer #4: Yes

4. Have the authors made all data underlying the findings in their manuscript fully available?

Reviewer #3: No

Reviewer #4: Yes

5. Is the manuscript presented in an intelligible fashion and written in standard English?

Reviewer #3: Yes

Reviewer #4: Yes

6. Review Comments to the Author

Reviewer #3: I read the manuscript again in detail, and it is well-structured and written well. However, the use of abbreviations in the abstract section is not recommended, and I recommend revising it.

Reviewer #4: (No Response)

7. PLOS authors have the option to publish the peer review history of their article (what does this mean?). If published, this will include your full peer review and any attached files.

Reviewer #3: **Yes: **Mesfin Abebe

Reviewer #4: **Yes: **Gossa Fetene Abebe

---

## [Editor Report · Acceptance letter]

13 Jan 2023

PONE-D-22-08128R3 

Exploring caregivers’ experiences of Kangaroo Mother Care in Bangladesh– a descriptive qualitative study 

Dear Dr. Sjömar:

I'm pleased to inform you that your manuscript has been deemed suitable for publication in PLOS ONE. Congratulations! Your manuscript is now with our production department. 

Kind regards, 

on behalf of

Dr. Tai-Heng Chen 

Academic Editor

PLOS ONE